# Drug-Induced Partial Immunosuppression for Preclinical Human Tumor Xenograft Models

**DOI:** 10.3390/cancers17244025

**Published:** 2025-12-17

**Authors:** Anton K. Gorbushin, Natalia A. Luzan, Victoriya D. Kakhanova, Anastasia A. Koshmanova, Daniil S. Grek, Ivan I. Voronkovskii, Vladislav M. Farniev, Elvira. S. Melikhova, Kirill A. Lukyanenko, Dmitriy V. Veprintsev, Evgeny V. Morozov, Maya A. Dymova, Elena V. Kuligina, Evgeny A. Pryakhin, Vladimir A. Richter, Elena V. Styazhkina, Ekaterina A. Lipetskaya, Tatiana A. Garkusha, Tatiana N. Zamay, Olga S. Kolovskaya, Andrey A. Narodov, Vadim V. Kumeiko, Maxim V. Berezovski, Anna S. Kichkailo

**Affiliations:** 1Therapeutic Faculty of Krasnoyarsk State Medical University Named After Prof. V.F. Voino-Yasenetsky, 660022 Krasnoyarsk, Russia; 2Krasnoyarsk Inter-District Ambulance Hospital Named After N.S. Karpovich, 660062 Krasnoyarsk, Russia; 3Laboratory for Biomolecular and Medical Technologies Krasnoyarsk State Medical University, 660022 Krasnoyarsk, Russia; 4Laboratory for Digital Controlled Drugs and Theranostics, Federal Research Center “Krasnoyarsk Science Center SB RAS”, 660036 Krasnoyarsk, Russia; 5School of Medicine and Life Sciences, Far Eastern Federal University, 690922 Vladivostok, Russia; 6Institute of Chemistry and Chemical Technology, Federal Research Center “Krasnoyarsk Science Center SB RAS”, 660036 Krasnoyarsk, Russia; 7Institute of Chemical Biology and Fundamental Medicine SB RAS, 630090 Novosibirsk, Russia; 8Southern Urals Federal Research and Clinical Center for Medical Biophysics (SUFRCC MB), 454141 Chelyabinsk, Russia; 9A.V. Zhirmunsky National Scientific Center of Marine Biology, Far Eastern Branch of Russian Academy of Sciences, 690041 Vladivostok, Russia; 10Department of Chemistry and Biomolecular Sciences, University of Ottawa, Ottawa, ON K1N6N5, Canada

**Keywords:** partial immunosuppression, xenograft model, cyclosporine, cyclophosphamide, orthotopic transplantation, glial brain tumors, lung cancer, breast cancer

## Abstract

Testing new cancer treatments often relies on animal models where human tumors can be grown. While genetically immunodeficient mice are the standard, they are costly and lack a functional immune system, which limits the study of tumor–immune interactions. To address this, we developed a novel protocol for pharmacological partial immunosuppression. This approach uses a short-term drug regimen to temporarily suppress the immune system of immunocompetent mice while crucially preserving its residual function. Unlike complete genetic immunodeficiency, our method maintains a partial immune microenvironment, enabling the study of malignant phenotypes in a more physiologically relevant context. We validated this model by successfully establishing xenografts of hard-to-treat cancers, including brain, lung, and breast tumors. Our protocol provides a reproducible, cost effective controlled and accessible tool for preclinical cancer research and drug development, offering a distinct advantage for studies that require a partially intact immune system.

## 1. Introduction

According to data from the International Agency for Research on Cancer (IARC), global cancer statistics record more than 10 million newly diagnosed cases annually, with deaths from malignant neoplasms reaching over 6 million per year. IARC’s predictive estimates indicate a significant increase in the cancer burden by 2030, with approximately 27 million new cancer cases and about 17 million cancer-related deaths projected annually [1]. In response to these epidemiological trends, modern research in medicinal chemistry is focused on the development of new therapeutic approaches, including (1) the design and synthesis of novel antitumor compounds [2]; (2) the isolation and pharmacological study of biologically active substances of natural origin with anticarcinogenic properties [3,4]; (3) the design of advanced delivery systems for chemotherapeutic agents [5]; and (4) the development of biological drugs, such as small interfering RNAs, short hairpin RNAs, and oncolytic viruses [6,7].

When evaluating the antitumor efficacy of new therapeutic compounds and delivery systems in vivo, the xenograft models in laboratory mice are predominant choice. This experimental system possesses significant predictive value, allowing researchers not only to analyze the test agents’ ability to inhibit tumor growth but also for the study of their impact on fundamental pathogenic mechanisms of carcinogenesis, including neoangiogenesis [8] and the formation of metastatic foci [9,10]. However, the translational value of these models depends critically on the biological relevance of the tumor microenvironment (TME), a key component of which is the host immune system.

Studies involving the creation of tumor xenografts traditionally use immunodeficient mice from several strains [11], whose development involved the sequential “knockout” of immune system components:Nude mice—A model with T-lymphocyte deficiency. A genetic characteristic of this strain is a mutation in the Foxn1 gene, leading to the absence of a functional thymus [12]. However, the preservation of B-lymphocytes and NK cells can limit the engraftment of certain tumor types [11,12].SCID mice—A model with severe combined immunodeficiency (absence of T- and B-lymphocytes) [13]. A mutation in the *Prkdc* gene disrupts DNA repair, preventing lymphocyte maturation. However, the preserved high NK cell activity and the potential for spontaneous immune recovery limit their use [11,12].NOD-SCID mice—An improved model where the SCID mutation is combined with the NOD (Non-Obese Diabetic) genetic background, which further weakens innate immunity [13]. Reduced NK cell activity, along with defects in macrophages and dendritic cells significantly improve the engraftment efficiency of xenografts [12].NSG/NOG mice are “third-generation” models with an additional deletion of the *Il2rg* gene. The complete absence of adaptive immunity (T- and B-cells) and severely impaired innate immunity (including NK cells) make them the most permissive models for engrafting human primary tumors and establishing humanized immune system models [12,14].

A critical limitation of these advanced genetic models is their complete immunodeficiency, which creates an artificial TME devoid of any functional immune components. This fundamentally restricts their utility for studying immunomodulating therapies and investigating how the immune contexture influences tumor biology and treatment response [15]. Beyond immunological constraints, PDX models face other significant challenges: variable and often low engraftment rates, a lengthy and costly generation process, and the issue of stromal replacement, where human stroma is supplanted by murine stroma, potentially altering signaling pathways and clonal evolution [15]. As an alternative, pharmacological immunosuppression in immunocompetent animals offers a pathway to create xenograft models while preserving a partially intact immune system. This approach is particularly valuable for studies requiring a more physiologically relevant TME, for short-term experiments, or when access to specialized immunodeficient strains is limited. The method typically involves cytostatic agents (such as cyclophosphamide) or sublethal irradiation to induce lymphocyte depletion [16,17,18]. However, a standardized protocol for achieving a defined state of partial immunosuppression is lacking, and comprehensive data on the dynamic changes in immune cell subpopulations following such interventions remain limited. In this context, we developed and characterized a combined pharmacological protocol for transient immunosuppression based on cyclosporine, cyclophosphamide, and ketoconazole, modifying an approach previously proposed by Jivrajani M. et al. [1]. The primary goal is not total immune ablation, but to create a transient and reproducible window of immunosuppression that is permissive for tumor engraftment while retaining residual immune landscape. The method has been successfully tested on models of glioblastoma, breast cancer, and non-small cell lung cancer [19,20,21].

This study aimed to comprehensively characterize a murine immunosuppression model based on an evaluation of the dynamics of key lymphocyte subpopulations (CD3^+^, CD3^+^CD4^+^, CD3^+^CD8^+^, CD19^+^) and the CD4/CD8 immunoregulatory index. The obtained data are intended to standardize conditions for reproducible xenotransplantation and the subsequent testing of antitumor drugs, providing a preclinical platform to investigate the role of a partially intact host immune system can be investigated.

## 2. Materials and Methods

### 2.1. Partial Immunosuppression

The study was conducted on six laboratory outbred male ICR (CD-1) mice, aged 4–6 weeks and weighing 20–25 g. The animals were housed in sterile individually ventilated cages under standard conditions. The mice were examined daily to assess their general condition, and body weight measurements were performed.

All housing, feeding, care, and euthanasia procedures were performed in accordance with the requirements of the “European Convention for the Protection of Vertebrate Animals used for Experimental and other Scientific Purposes” (Strasbourg, 1986) [22] and the “Rules for Conducting Work with the Use of Experimental Animals” (Order 755 of 12 August 1977 of the USSR Ministry of Health) [23]. The animal study protocol was approved by the Local Ethics Committee of Krasnoyarsk State Medical University in Krasnoyarsk, Russia (#115/2022 dated 28 November 2022).

A combined immunosuppressive regimen using three drugs was applied to suppress the immune response in the mice: (1) cyclosporine (Catalent, Eberbach, Germany) at 20 mg/kg intraperitoneally every 48 h for 12 days; (2) cyclophosphamide (Baxter Oncology, Halle (Westfalen), GmbH, Germany) at 60 mg/kg intraperitoneally every 48 h for 8 days (administration was discontinued after four injections due to its cytotoxicity to all actively dividing cells and because tumor xenotransplantation was scheduled for day 10); (3) ketoconazole (ZiO-Health, Russia) at 10 mg/kg orally, provided ad libitum in drinking water for the entire 12-day immunosuppression period (Figure 1).Throughout the experiment, body weight was recorded, and body temperature was monitored via pre-implanted subcutaneous temperature microchips (Animal Guard’s Bio-Thermo™ Microchips, Rahway, NJ, USA) with the reader (Global Pocket Reader™ Plus for Horses—Merck Animal Health, Rahway, NJ, USA).

Blood samples were collected from the peripheral tail veins on days 1, 5, 8, 12, 16, and 21 following the initiation of immunosuppression, as previously described [24]. Briefly, each mouse was restrained, and its tail was immersed in warm water (~40 °C) for 30 s to promote vasodilation. A rubber tourniquet was then applied at the base of the tail, and the tail was disinfected. Using a pre-heparinized syringe with a 27 G needle, the tail vein was punctured, the tourniquet was released, and 40–50 µL of blood was collected. The puncture site was treated with a disinfectant. It is important to note that the mice in this immunomonitoring experiment did not receive tumor cell inoculations; they were subjected solely to the immunosuppressive protocol to characterize its specific effects on the immune system. Tumor transplantation studies were performed in separate cohorts of animals.

### 2.2. Sample Preparation

Blood samples were immediately transferred into heparinized microtubes and gently mixed. The blood was centrifuged at 3500 rpm for 3 min, after which the plasma was removed. A lysing solution (0.42% NH_4_Cl, Tris HCl, heparin) was added to the pellet at a 1:5 (pellet/lyse) ratio and mixed thoroughly. The samples were incubated for 15 min at room temperature, followed by centrifugation at 3500 rpm for 3 min and removal of the supernatant. This procedure was repeated twice. Subsequently, the cell pellet was washed with phosphate-buffered saline (PBS, pH 7.4). For immunostaining staining, each sample was divided into 3 aliquots and stained with fluorescently labeled antibodies according to the manufacturer’s instructions (Cloud-Clone Corp, Wuhan, China).

### 2.3. Flow Cytometry

The subpopulational composition of peripheral blood lymphocytes was analyzed using a Cytomics FC500 flow cytometer (Beckman Coulter, Indianapolis, IN, USA). The following commercially available fluorescently labeled antibodies (Cloud-Clone Corp., Katy, TX, USA) were used: APC-conjugated anti CD19 (Cat. No. B324740, Lot 115546), FITC-conjugated anti CD8 (Cat. No. B312598, Lot 100722), PE-conjugated anti CD3 (Cat. No. B321239, Lot 100236), FITC-conjugated anti CD4 (Cat. No. B336799, Lot 100540) from Cloud-Clone Corp. (USA). Subsequent data analysis, including the implementation of the gating strategy and quantification of cell populations, was performed using Kaluza Analysis Software (version 2.1, Beckman Coulter). For immunophenotypic analysis, each blood sample was divided into three aliquots and stained with the following antibody panels:

Panel 1: PE-conjugated anti-CD3 (Cat. No. B321239) + FITC-conjugated anti-CD4 (Cat. No. B336799) + APC-conjugated anti-CD19 (Cat. No. B324740).

Panel 2: PE-conjugated anti-CD3 (Cat. No. B321239) + FITC-conjugated anti-CD8 (Cat. No. B312598) + APC-conjugated anti-CD19 (Cat. No. B324740).

Panel 3 (Control): Appropriate fluorescence-minus-one (FMO) or isotype control stains.

The gating strategy is illustrated in Appendix A.

Briefly, lymphocytes were first identified and gated based on their characteristic forward scatter (FSC) and side scatter (SSC) properties. Within this lymphocyte gate, a minimum of 5000 events were collected per sample. The following major lymphocyte subpopulations were quantified: total T-lymphocytes (CD3^+^), T-helper cells (CD3^+^CD4^+^), cytotoxic T-lymphocytes (CD3^+^CD8^+^), and B-lymphocytes (CD19^+^). Finally, the immunoregulatory CD4^+^/CD8^+^ T-cell ratio was calculated from the obtained frequencies of the respective subpopulations.

### 2.4. Statistical Analysis

Statistical analysis was performed using the open-source Anaconda package (Python 3.12.4).

Data distributions are described using medians and quartiles, which are presented in all graphs in Section 3.1. Group comparisons were performed using the non-parametric Mann–Whitney U test. A significance level (alpha) of 0.05 was used.

To account for multiple hypothesis testing, the Bonferroni correction was applied. The sample size (n = 6 mice per group) is specified at the beginning of the section.

### 2.5. Patient-Derived Tumor Samples

All procedures involving human biological samples were conducted in accordance with the ethical principles of the Declaration of Helsinki. The study protocols were reviewed and approved by the following Local Ethics Committees:

The Local Ethics Committee of the Krasnoyarsk Inter-District Ambulance Hospital named after N.S. Karpovich (protocol code #20/11/2016, date of approval: 20 November 2016).

The Local Ethics Committee of Krasnoyarsk State Medical University in Krasnoyarsk, Russia (protocol code #37/2012, date of approval: 31 January 2012).

Human brain glioma samples were obtained from patients undergoing complete curative resection at the N.S. Karpovich Krasnoyarsk Inter-District Ambulance Hospital. Non-small-cell lung cancer (NSCLC) samples were procured from the A.I. Kryzhanovsky Krasnoyarsk Regional Clinical Oncology Dispensary. Written informed consent was obtained from all participants prior to sample collection.

Immediately following surgical resection, solid tumor tissues were aseptically excised and placed in ice-cold Dulbecco’s Modified Eagle Medium (DMEM), containing 1000 U/mL penicillin G and 1000 µg/mL streptomycin. Samples were maintained on ice and delivered to the laboratory within a post-resection window 2 to 4 h.

### 2.6. Primary Cell Isolation and Culture

Primary human brain glioma and non-small cell lung cancer cultures were established from freshly resected tumor tissue. All subsequent procedures were performed under aseptic conditions in a laminar flow hood. Upon arrival, the transport medium was aspirated, and the tissue fragments were washed three times with 5 mL of cold DPBS to remove blood residues. The tissue was then transferred to a sterile Petri dish containing 1–2 mL of cold DPBS. Necrotic areas and blood clots were meticulously removed using sterile forceps and a scalpel.

The remaining viable tissue was minced into fragments approximately 1 mm^3^. For glial tumors, fragments were placed directly into culture flasks containing complete growth medium to facilitate spheroid formation. To establish adherent monolayer cultures, the minced tissue was dissociated and filtered through a 70-μm cell strainer into a sterile 15 mL conical tube. The filtrate was washed twice with DPBS via centrifugation (2000 rpm for 5 min).

The resulting cell pellet was resuspended in 2 mL of DPBS and carefully layered over 3 mL of Lymphocyte Separation Medium. Following centrifugation (1800 rpm for 10 min), the buffy coat interface, containing the mononuclear cells, was collected and transferred to a new tube with DPBS. A final centrifugation step (2000 rpm for 5 min) was performed to pellet the cells, which were then resuspended in nutrient medium and seeded into culture flasks. Cells were maintained in a humidified incubator at 37 °C in an atmosphere of 5% CO_2_. The medium was changed two to three times a week.

Culture medium for brain glioma cells: DMEM/F-12 (1:1) mixture supplemented with 10% fetal bovine serum (FBS), 2 mM L-glutamine, and 100 U/mL penicillin–100 µg/mL streptomycin.

Culture medium for lung cancer (NSCLC) cells: High-glucose DMEM supplemented with 10% FBS, 2 mM L-glutamine, 100 U/mL penicillin–100 µg/mL streptomycin, 20 µg/mL insulin, 10 µg/mL transferrin, 25 nM sodium selenite (ITS supplement), and 1 ng/mL epidermal growth factor (EGF).

For passaging, the culture medium was aspirated, and the cells were rinsed with DPBS without Ca^2+^ and Mg^2+^. To detach the cells, 3–5 mL of Ca^2+^/Mg^2+^-free DPBS was added, and the flask was incubated for 2–3 min at room temperature. The cells were then gently dislodged by pipetting. For complete dissociation, 2–4 mL of Versene solution was added, followed by a 5–15 incubation. The resulting cell suspension was centrifuged at 2500 rpm for 3 min. The supernatant was discarded, and the pellet was resuspended in fresh complete medium for reseeding.

### 2.7. Preparation of Cell Spheroids

Cell spheroids were prepared using standard, widely used technique [25]. A sterile, molten 1% agarose solution was added to the wells of a sterile 96-well V-bottom plate (100 µL per well), and the excess was immediately removed. The plates were incubated at room temperature for 15 min to allow the agarose coating to polymerize. A suspension of U87MG or primary glial tumor cells was seeded into the prepared plate wells (100 µL per well, at a final concentration of 4 × 10^3^ cells per well). The plates were cultured at 37 °C and 5% CO_2_ in a humidified incubator. Spheroid formation was monitored daily using an inverted phase-contrast microscope or a stereomicroscope; U87MG cells typically formed compact, rounded aggregates within 24–48 h. On the day of transplantation, spheroids were stained with the vital dye Calcein-AM (Sigma-Aldrich, St. Louis, MO, USA) at a working concentration of 1 µM in the culture medium for 1 h.

Subsequently, the stained spheroids were washed with PBS maintained in fresh culture medium at 37 °C until implantation.

### 2.8. Anesthesia of Laboratory Animals Prior to Surgery

Anesthesia was induced via intraperitoneal injection of a mixture of “Zoletil 100” (tiletamine/zolazepam, Virbac, France) at a dose of 20 mg/kg body weight and Rometar 2% (xylazine hydrochloride, Bioveta, Ivanovice na Hané, Czech Republic) at a dose of 16 mg/kg body weight. Following injection, the animals were closely monitored. The depth of anesthesia was assessed by the suppression of corneal and palpebral reflexes. If these reflexes remained present 10 min post-injection, a single supplemental intraperitoneal dose of the anesthetic mixture, equivalent to one-third of the initial dose volume, was administered.

### 2.9. Orthotopic Glioma Spheroid Transplantation and Cranial Window Implantation in Mice

Forty mice were used in the experiment: 20 were CBA mice and 20 were C57BL/6. Each strain was divided into two groups: 10 control mice without immunosuppression and 10 mice subjected to immunosuppression using the protocol described above.

Prior to trepanation, the hair on the head was shaved under the anesthesia. The skin surface was treated with a 20% aqueous ethanol solution. Subsequently, a skin incision was made in the frontal region of the mouse’s head. To ensure reproducible results, the burr hole was created and the tumor was induced at the same stereotaxic coordinate in the cerebral cortex.

Immediately before creating the burr hole, after the skin incision and exposure of the skull, the skull was fixed in a stereotaxic apparatus. The skull was marked using a pen held in the instrument clamp. The intersection of the sagittal and sutures, known as the lambda point, served as the zero coordinate. Using the screws of the stereotaxic apparatus, the marker was aligned with the lambda point. The pen was then moved 3 mm anterior (rostral) along the sagittal suture and 4 mm lateral (from the midline) using the apparatus screws. The marker was then lowered via a screw until it touched the skull, leaving a mark at the center of the future hole.

The mouse was removed from the stereotaxic apparatus and placed on a flat surface covered with a wipe for trepanation. A burr hole approximately 4–6 mm in diameter was created at the pre-marked site using a Dremel 4250 (Dremel, Mexico) rotary tool fitted with an inverted cone-shaped, sintered diamond dental bur (G8006 NTI) with a 5 mm working surface diameter. Drilling was performed until the parietal bone was thinned but not completely penetrated, carefully avoiding the intersection of the sagittal and coronal sutures by slightly diverting the bur laterally (Figure 1). The drilling site and the animal’s eyes were periodically irrigated with cool saline solution to prevent significant brain overheating and corneal drying, respectively.

A thin remnant of bone was lifted using a curved metal probe, gently guided along the cut, and separated from the meninges with forceps. After trepanation, the mice were immediately repositioned in the stereotaxic apparatus. Using the apparatus screws, the needle of a syringe was aligned with the lambda point and then moved 3 mm in the sagittal direction towards the nose and 4 mm in the frontal direction towards the ear. A precision syringe (Reno, Nevada Hamilton, NV, USA) was lowered via the vertical screw to a depth of 300 µm below the cortical surface. The mouse was then moved under a binocular stereomicroscope, and under visual guidance, a single spheroid was implanted inside the cortical tract using a G21 needle syringe.

A cranial window was formed. A dental micro-hybrid composite restorative material (TE-Econom Plus, Ivoclar, Ellwangen, Germany), was applied to secure a coverslip (5–7 mm in diameter). The coverslip was placed onto the composite, and a raised border was formed around it to retain a drop of fluid above the glass.

### 2.10. Orthotopic Transplantation of Glioma Cell Cultures Through a Micro-Perforation

Fifteen ICR mice participated in the experiment. Each strain was divided into two groups: 5 control mice without immunosuppression and 10 mice subjected to immunosuppression.

The procedure was identical to that used for transplantation through a cranial window, with the following modifications: a micro-perforation was created using either a 1 mm electric trephine or by manually rotating a syringe needle. A single 0.5 mm spheroid together with 2.5 × 10^5^ dissociated tumor/stromal cells in 10 µL of a 1:1 GrowDex/DMEM hydrogel medium were loaded into a Hamilton syringe. This cell suspension was flanked by two 2 µL plugs of a denser 2:1 GrowDex/DMEM hydrogel. The tumor cells were then inoculated into the mouse brain through this perforation, after which the skin incision was sutured.

### 2.11. Intravital Fluorescent Microscopy Through a Transcranial Optical Window

Before in vivo imaging, animals were anesthetized and immobilized in a holder. A droplet of distilled water was applied to the surface of the cranial window to create an optical interface with the cerebrospinal fluid below and to allow the use of a water-immersion objective. Intravital fluorescence microscopy was performed using an FV1200 (Olympus, Tokyo, Japan) laser scanning confocal microscope with a 10× water-immersion objective in confocal Z-stack mode. All images were acquired using consistent laser power and detector sensitivity settings, without software-based signal amplification or noise reduction. These parameters were initially optimized based on the fluorescence intensity observed during the first acquisition to ensure a detectable intensity gradient across the image. Two-dimensional maximum intensity projections of Z-stacks were generated using FV10-ASW software (version 4.1, Olympus). Three-dimensional surface reconstructions were rendered from the Z-stacks using the Imaris software package (version 7.6.5, Bitplane).

### 2.12. Orthotopic Xenotransplantation of a Human Lung Cancer Cells into Mice with Drug-Induced Immunosuppression

A human non-small cell lung cancer (NSCLC) xenograft model was established by orthotopic implantation of a primary patient-derived cell culture into the right lung. A monodisperse cell suspension (containing 3 × 10^5^ cells in 6 µL of a 1:1 mixture of GrowDex™ and DMEM) was injected into the intercostal space of ten immunosuppressed ICR mice using a 20 µL Hamilton syringe.

Tumor growth was monitored at 3 weeks post-transplantation using MRI with contrast enhancement by gadodiamide (Omniscan, GE Healthcare Ireland Limited, Carrigtohill, Ireland) and by histological examination of autopsy samples.

### 2.13. Xenotransplantation of a Breast Cancer Cell Line into Mice with Drug-Induced Immunosuppression

A human breast cancer model was established by subcutaneous injection of the MCF7 breast cancer cell line into the axillary region of 10 ICR immunosuppressed mice. The MCF7 cell line was obtained from the Russian Cell Culture Collection (Institute of Cytology, Russian Academy of Sciences, Saint Peterburg, Russia). A monodisperse cell suspension containing 5 × 10^5^ cells in 10 µL of DMEM supplemented with 10% FBS, 100 IU/mL penicillin G, 100 mg/L streptomycin, and 2 mM L-glutamine was administered using a U50 syringe (Vogt Medical GmbH, Karlsruhe, Germany). Tumor engraftment and growth were assessed 3 weeks post-transplantation by histological analyses of autopsy samples.

### 2.14. MRI

MRI was performed according to a modified protocol described previously [19,26]. Briefly, anesthetized and immobilized mice were imaged using an Avance DPX 200 spectrometer (Bruker BioSpin GmbH, Rheinstetten, Germany) using the contrast agent gadodiamide (Omniscan, GE Healthcare Ireland Limited, Ireland). Two-dimensional slice-selective images were acquired using a multi-slice multi-echo (MSME) sequence implemented in the Paravision 4.0 software (Bruker BioSpin, Germany). Acquisition parameters were as follows: slice thickness, 0.71 mm; field of view (FOV), 40 mm; matrix size, 256 × 256; in-plane spatial resolution, 156 × 156 µm^2^; repetition time (TR), 600 ms; echo time (TE), 4.7 ms, providing T1-weighted contrast. The total acquisition time was 10 min.

## 3. Results

### 3.1. The Impact of Partial Drug-Induced Immunosuppression on Key Lymphocyte Subpopulations

Following the first injection of cyclosporine and cyclophosphamide, all treated mice developed a characteristic clinical syndrome comprising reduced general activity, lethargy, poor coat quality, and decreased food and water intake. These signs peaked in severity by day 5 post-initiation of immunosuppression. A gradual recovery in behavior, consumption rates, and coat condition was observed after the final cyclophosphamide injection (day 8). Body temperature showed a corresponding decline, decreasing from a baseline mean of 37.6 °C to 37.2 °C by day 5, followed by a rebound, rising to 38.0 °C by day 16 and reaching 38.6 °C by day 21 (Appendix A). Similar dynamics were observed for body weight: a loss was recorded by day 5, followed by a positive trend of weight gain, indicating the onset of recovery and systemic compensation. Considerable inter-individual variability in body weight was noted during the immunosuppressive period (Figure 2A). Notably, no mortality occurred throughout the experiment. The observed hypothermia during the treatment phase likely reflects drug-induced toxicity and general physiological depression, while the subsequent rebound hyperthermia may indicate an inflammatory response or a compensatory metabolic activation during recovery.

It is known that combined immunosuppression with cyclosporine, cyclophosphamide, and ketoconazole reduces the total number of leukocytes and lymphocytes in animals [1]. Cyclosporine is a highly effective immunosuppressant whose mechanism of action is based on the selective inhibition of T-lymphocyte activation. By binding to the intracellular receptor cyclophilin, cyclosporine forms a complex that suppresses the activity of the enzyme calcineurin. Inhibition of calcineurin blocks the translocation of the Nuclear Factor of Activated T-cells (NFAT) into the nucleus, which is normally necessary for the transcription of the interleukin-2 (IL-2) gene. Reduced expression of IL-2, a key cytokine for the T-cell response, leads to the suppression of clonal proliferation and differentiation of T-lymphocytes [27,28]. Ketoconazole inhibits the metabolism of cyclosporine by suppressing the activity of cytochrome P450, leading to an increased plasma concentration of cyclosporine, prolonging its action, and enhancing its therapeutic effect [29,30,31]. Additionally, ketoconazole provides prophylaxis against opportunistic mycoses, the risk of which significantly increases during immunosuppressive therapy. The action of cyclophosphamide is based on its ability to alkylate DNA, thereby blocking its replication and transcription. This drug is cytotoxic and induces profound immunosuppression, manifested as a significant reduction in the pools of neutrophils, B- and T-cells, as well as natural killer (NK) cells [32].

The CD3 marker, being a key component of the T-cell receptor (TCR), is expressed on all mature T-lymphocytes and serves as the primary parameter for assessing the cellular arm of adaptive immunity. In our experiment, combined immunosuppression led to a statistically significant decrease in the relative count of CD3^+^ lymphocytes by day 8, followed by its restoration to the baseline level by day 12 (Figure 2B).

The dynamics of the relative counts of both T-helper (CD3^+^CD4^+^) and T-cytotoxic (CD3^+^CD8^+^) cells generally mirrored the changes in the total T-lymphocyte (CD3^+^) population; however, the differences did not reach statistical significance. Notably, the median level of T-helper cells, having decreased by day 8 (the day of the final drug administration), did not recover to the initial level by the end of the experiment (Figure 3A). In contrast, the median level of T-cytotoxic cells, after reaching its minimum on day 8, demonstrated significant growth following drug withdrawal and exceeded the baseline values by day 21 (Figure 3B).

The dynamics of the relative CD4^+^ (Figure 3C) and CD8^+^ (Figure 3D) cell counts correlated with the changes in the CD3^+^CD4^+^ and CD3^+^CD8^+^ populations, respectively.

The slow and incomplete recovery of the CD4^+^ T-helper cell pool after therapy cessation is apparently associated with their increased sensitivity to apoptosis and dependence on the constant supply of naive cells from the thymus. In our experiment, thymic function was likely significantly suppressed by cyclophosphamide, leading to a prolonged depletion of the naive CD4^+^ pool and its slow reconstitution.

In contrast, the rapid recovery of T-cytotoxic lymphocytes (CD8^+^) to levels exceeding baseline after the cessation of immunosuppression is due to their high proliferative potential and independence from thymic output. The CD8^+^ T-cell pool is largely maintained through peripheral proliferation, which can be extremely active after cyclosporine withdrawal (day 8). In our case, a “rebound” phenomenon is observed precisely on day 12. Furthermore, CD8^+^ lymphocytes are more resistant to apoptosis. After immunosuppression is discontinued, the remaining CD8^+^ cell clones actively proliferate, restoring their pool to a level exceeding the baseline (by day 21), which likely represents a compensatory response to the experienced immunosuppression.

The immunoregulatory index, calculated as the CD4^+^/CD8^+^ cell ratio, is an important parameter reflecting the state of the immune system and the balance between immune response activation and suppression. In our experiment, starting from day 8, a pronounced and progressive decrease in this index was observed, which continued even after the cessation of immunosuppressive therapy (Figure 4A). The sharp and significant drop in the CD4^+^/CD8^+^ index confirms the high efficacy of the combined therapy with cyclosporine and cyclophosphamide, which exerted a powerful suppressive effect on the immune system.

CD19 is a highly specific membrane marker expressed on B-lymphocytes at all stages of differentiation, except for terminally differentiated plasma cells. As a key component of the co-receptor complex, this antigen plays a critical role in the development, activation, and functional regulation of humoral immunity cells. Consequently, the quantitative assessment of CD19 levels and the determination of the percentage of CD19^+^ lymphocytes in peripheral blood serve as informative diagnostic parameters for evaluating the status of the humoral arm.

In our study, the relative count of B-lymphocytes began to decrease after the first administration of cyclosporine and cyclophosphamide, reaching a statistically significant minimum by day 8 of the experiment. However, by day 12, a sharp increase in the level of CD19^+^ cells was observed, followed by a gradual decline (Figure 4B). The minimum values were recorded on day 8—the day of the final cyclophosphamide administration. This effect is likely due to the cytotoxic action of cyclophosphamide on the B-cell compartment. The drug affects rapidly proliferating cells, which include B-lymphocytes, inducing a biphasic response: (1) a depression phase and (2) a recovery (rebound) phase, which can be followed by a compensatory decline. Thus, by day 8, the pool of mature peripheral B-lymphocytes is nearly depleted, and their bone marrow hematopoiesis is suppressed. After cyclophosphamide withdrawal (after day 8), the inhibitory effect on the bone marrow and lymphoid precursors ceases, while the continued administration of cyclosporine, which does not suppress hematopoiesis, allows the observation of the rebound phase [33]. The body responds to profound leukopenia by releasing hematopoietic growth factors and activating progenitor cells. The bone marrow initiates hypercompensation, “catching up,” leading to a massive release of immature and mature B-lymphocytes into the peripheral blood. The subsequent decline represents the normalization and redistribution phase. After the initial regeneration “surge,” the hematopoietic system begins to return to homeostasis, and the rate of progenitor cell proliferation and differentiation decreases to the basal level. It is important to note that cyclosporine, unlike cyclophosphamide, continues to act (day 12 from the experiment start is the last day of cyclosporine administration). It selectively inhibits T-lymphocyte activation, and T-cells, in turn, are key helpers for the final maturation and functioning of B-cells. This correlates with the obtained data on the cellular immune arm described above. Thus, cyclosporine may hinder the complete recovery and stabilization of the normal B-lymphocyte pool, causing the subsequent gradual decline after the rebound phase.

### 3.2. Human Glioma Model on Mice with Partial Drug-Induced Immunosuppression

Tumor models were established via orthotopic transplantation of glioma cell cultures or tumor spheroids through a cortical micro-perforation or trepanation hole with the installation of a cranial window (Figure 5). The figure shows the point for the proposed spheroid transplantation; the use of coordinates and stereotactic apparatus improves reproducibility and repeatability.

This approach successfully induced intracranial tumors suitable for long-term monitoring. When a cranial window was utilized, tumor growth could be visualized in vivo through the transparent glass implant using confocal microscopy. In vivo microscopic visualization of the transplanted tumor was performed using a benchtop FV1200-MPE laser scanning microscope (Olympus, Japan), with the data presented in Figure 6. This image demonstrates tumor development within mouse brain tissue (cortex) after spheroid transplantation. Over time, the bioconvertible cellular tracer Calcein-AM is released from the cells, and by day 7, fluorescence intensity significantly decreases. However, the image demonstrates an increase in cell number and tumor volume. Using IMARIS x64 7.6.5 software, it was determined that the cell count doubled between days 3 and 7.

Tumor growth was assessed on day 14 post-transplantation via contrast-enhanced MRI using the gadolinium-based contrast agent gadodiamide Omniscan (GE Healthcare Ireland Limited, Ireland) and histological analysis of autopsy specimens (Figure 7). For MRI, a representative image of a mouse with xenografted glioma (Figure 7(A1)) is shown alongside a scan from non-immunosuppressed mouse (Figure 7(A2)), and control intact mouse (Figure 7(A3)). Tumor presence was further confirmed by post-mortem analysis (Figure 8B) and histological analyses (Figure 7C). Upon microscopic examination of the mouse brain sample in the implantation area, medium-sized, process-bearing tumor cells with mildly basophilic and indistinct cytoplasm were identified. The nuclei of these cells were polymorphic and contained prominent nucleoli. Within the branched matrix between the tumor cells, lymphocytes with mature morphology were observed, located within small lacunae. This histological picture, characterized by the process-bearing tumor cells with nuclear polymorphism and the lymphoid infiltration in the stroma, was consistent with the growth of a malignant glial tumor (Figure 7C).

### 3.3. Human Lung Cancer Model on Mice with Partial Drug-Induced Immunosuppression

We successfully established a patient-derived orthotopic xenograft model of human lung cancer on 10 mice with drug-induced immunosuppression (Figure 8). The tumors reliably engrafted and exhibited progressive growth, becoming clearly detectable via non-invasive magnetic resonance imaging (MRI) following administration of gadodiamide (Omniscan). A representative MRI scan of a mouse bearing a lung cancer xenograft (Figure 8(A1)) is shown alongside a scan from non-immunosuppressed mouse (Figure 8(A2)), and control intact mouse (Figure 8(A3)). Tumor engraftment and growth were conclusively confirmed by post-mortem histological analysis as early as three weeks post-inoculation (Figure 8B). Within the pulmonary tissue, aggregates of tumor cells were identified (Figure 8C). These cells possessed medium-sized, oval nuclei and moderately eosinophilic cytoplasm. The area of tumor growth also contained lymphocytes with mature morphology, consistent with small lymphocytes (Figure 8C). This model provides a robust tool for studies on lung tumor therapy and visualization. The overall tumor engraftment rate using this pharmacological protocol was approximately 80%, with successful establishment showing significant variation across different tumor types and, notably, between individual patient-derived samples within the same tumor type. Among the tested models, engraftment was most consistent and efficient for carcinomas of pulmonary origin.

### 3.4. Breast Cancer Cell Line Model on Mice with Partial Drug-Induced Immunosuppression

We successfully demonstrated that our model of drug-induced immunosuppression supports the reliable growth of subcutaneously transplanted cancer cell lines. As a representative example, the MCF7 breast cancer cells exhibited robust engraftment and formed solid tumors in the axillary region. Histological examination of samples collected three weeks post-transplantation confirmed the presence of viable tumor tissue (Figure 9). These findings establish the suitability of this model for preclinical research, including investigations into tumor biology and therapeutic efficacy. Upon examination of subcutaneously transplanted tumor samples obtained from mice using immunohistochemistry with an antibody to Her2/neu, tumor cell complexes exhibiting intense, diffuse membranous expression of the Her2/neu antibody (3+++) were visualized.

## 4. Discussion

The development of robust and clinically relevant preclinical models is a cornerstone of translational oncology. This study successfully characterized and validated a combined pharmacological immunosuppression protocol as a viable and effective alternative to genetically immunodeficient mice for generating human tumor xenografts. Our findings confirm the central hypothesis that a regimen of cyclosporine, cyclophosphamide, and ketoconazole can induce a profound yet partial state of immunodeficiency in outbred ICR mice, sufficient for the engraftment and growth of various human cancers.

The immunological monitoring conducted in this study provides a critical, data-driven foundation for the model. The significant yet reversible reduction in CD3^+^ T-cells and CD19^+^ B-cells by day 8 of the protocol aligns with the known mechanisms of action of the drugs used. Cyclosporine effectively depletes proliferating lymphocyte pools [27,28], while cyclophosphamide blocking DNA replication and transcription [32]. The synergistic role of ketoconazole, which inhibits cyclosporine metabolism and provides antifungal prophylaxis, was crucial for both the protocol’s efficacy and animal welfare [29,30,31].

Importantly, these profound immunological shifts were mirrored by systemic physiological changes. Alongside the expected lethargy and weight loss, we recorded a significant drop in core body temperature, from a baseline of 37.6 °C to 37.2 °C by day 5, coinciding with the peak of leukopenia. This hypothermia likely reflects a state of reduced metabolic activity and diminished cytokine-driven thermogenesis due to immunosuppression. Following drug withdrawal, the recovery of immune cell counts was accompanied by a rebound hyperthermia, with temperatures rising to 38.0 °C and 38.6 °C on days 16 and 21, respectively, indicative of a systemic inflammatory response and heightened metabolic activity during immune reconstitution.

The most striking immunological finding was the profound and progressive decrease in the CD4/CD8 index. This shift, driven by the rapid recovery of CD8^+^ T-cells surpassing baseline levels and the sluggish reconstitution of CD4^+^ T-helper cells, creates a uniquely permissive environment for xenograft acceptance. The CD4/CD8 imbalance indicates a disrupted immunoregulatory balance, which is a hallmark of many pathological states and is effectively exploited here to facilitate tumor engraftment. The biphasic “depression and rebound” dynamics observed in the B-cell population further underscore the potent, yet transient and controllable, nature of the immunosuppression, which is a key safety feature of the protocol.

Our results are in strong agreement with the foundational work of Jivrajani et al. [1], who first proposed this combination. However, our study significantly expands upon it by providing a comprehensive, longitudinal characterization of key lymphocyte subpopulations, thereby moving beyond simple leukocyte counts to offer a more nuanced understanding of the immune landscape during and after treatment. This detailed immunological profiling addresses a significant gap in the literature concerning pharmacological immunosuppression protocols, where the precise state of the immune system often remains a “black box.”

The true validation of any immunosuppression model lies in its functional utility for xenotransplantation. Our successful establishment of orthotopic glioma and lung cancer models, as well as a subcutaneous breast cancer model, demonstrates the broad applicability and robustness of this protocol. The orthotopic models, in particular, are noteworthy. The faithful growth of gliomas in the brain microenvironment, confirmed by in vivo microscopy and MRI, and the engraftment of lung cancer in the pulmonary niche, highlight the protocol’s suitability for studying tumors in their anatomically correct context. This is a significant advantage, as the organ-specific microenvironment is known to critically influence tumor biology, metastatic potential, and drug response [8,9]. The use of patient-derived primary cultures for glioma and lung cancer further enhances the clinical relevance of our models, as they better preserve the genomic and phenotypic heterogeneity of the original tumors compared to immortalized cell lines.

When placed in the broadest context of cancer research, this pharmacological model offers several compelling advantages over traditional genetically engineered immunodeficient strains. First, it is markedly more cost-effective and accessible, as it utilizes readily available outbred mice, lowering the barrier to entry for many laboratories. Second, its flexibility is a major strength. The duration and intensity of immuno-suppression can be theoretically adjusted to suit the needs of specific experiments, such as short-term drug efficacy studies or longer-term investigations into metastatic spread. Third, and perhaps most important, this model establishes a state of partial immunosuppression, thereby preserving a residual immune landscape. This is crucial for preclinical studies of immunotherapies, oncolytic viruses, and other treatment modalities whose efficacy is intrinsically linked to immune interactions [7]. In contrast, the completely “immune-null” environment of NSG mice may not accurately recapitulate the complex interplay between a therapy, the tumor, and a partially functional immune system, as is often the case in human patients.

Despite its strengths, certain limitations of this approach must be acknowledged. The transient nature of the immunosuppression may not be suitable for studies re-quiring very long-term tumor observation. Furthermore, while we characterized systemic immune populations, the specific composition and role of intratumoral immune cells in these models remain to be fully elucidated. The potential systemic toxicity of the drug regimen, though manageable in our study, requires careful monitoring.

Future research directions emerging from this work are multifaceted. Firstly, the protocol should be tested with a wider array of human cancers, including those known to be difficult to engraft, such as prostate and pancreatic cancers. Secondly, it would be highly informative to “humanize” these pharmacologically immunosuppressed mice by co-engrafting human immune cells, creating a chimeric model that allows for the study of human-specific tumor–immune interactions in a more controlled and affordable system than is possible with NSG mice. Finally, integrating this xenograft platform with the testing of novel therapeutics, particularly immunotherapies and targeted agents, will be the ultimate test of its predictive value for clinical success.

The combined immunosuppression protocol presented here is not merely a substitute for genetically immunodeficient models but a versatile and powerful tool in its own right. By providing a detailed immunological characterization and demonstrating its efficacy across multiple orthotopic tumor models, we have established a standardized, reproducible, and cost-effective platform that explicitly incorporates a partially intact immune system. This model holds significant promise for accelerating preclinical cancer research and the development of next-generation anticancer therapies.

## 5. Conclusions

Our study confirmed that the developed pharmacological immunosuppression protocol based on cyclosporine, cyclophosphamide, and ketoconazole enables the creation of a reproducible immunodeficiency model in outbred mice. The protocol ensures profound and controlled suppression of both cellular and humoral immunity, as objectively demonstrated by the marked dynamics of key lymphocyte subpopulations (CD3^+^, CD4^+^, CD8^+^, CD19^+^).

It was established that the components of the regimen act synergistically: cyclophosphamide induces direct cytotoxicity and depletion of the proliferating lymphocyte pool, while cyclosporine, potentiated by ketoconazole, provides a prolonged functional blockade of T-cell activation.

The proposed model offers several advantages over the use of congenitally immunodeficient strains, such as nude mice. This includes greater flexibility, cost-effectiveness, and the unique capacity to model the dynamic “tumor-immune system” interactions within a partially suppressed but intact host. This feature is particularly important for the preclinical evaluation of novel immunotherapeutic approaches, oncolytic virotherapy, or monoclonal antibody therapy.

The high efficacy of this protocol was demonstrated by the successful and reproducible engraftment of orthotopic and subcutaneous xenografts derived from various human tumors, including glioma, lung, and breast cancer.

These findings show that the restoration of the mice’s immune status, as evidenced by blood immunocytochemistry and histology, coincides with progressive tumor growth and a functional tumor microenvironment. Thus, this protocol can be recommended as a valuable and accessible tool in preclinical research for testing the efficacy of new antitumor drugs and their delivery systems, as well as for investigating the fundamental mechanisms of carcinogenesis and metastasis under conditions of controlled and reversible immunosuppression. The high efficacy of this protocol was further confirmed by the successful and reproducible engraftment of orthotopic and subcutaneous xenografts derived from various human tumors, including glioma, lung, and breast cancer.

## Figures and Tables

**Figure 1 cancers-17-04025-f001:**
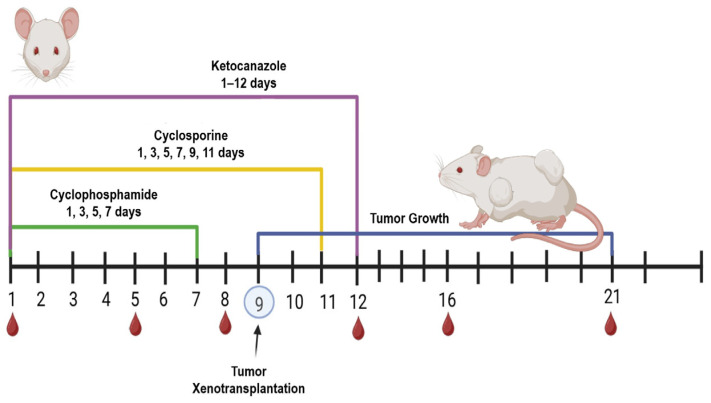
Schematic of the drug-induced partial immunosuppression protocol in immunocompetent mice.

**Figure 2 cancers-17-04025-f002:**
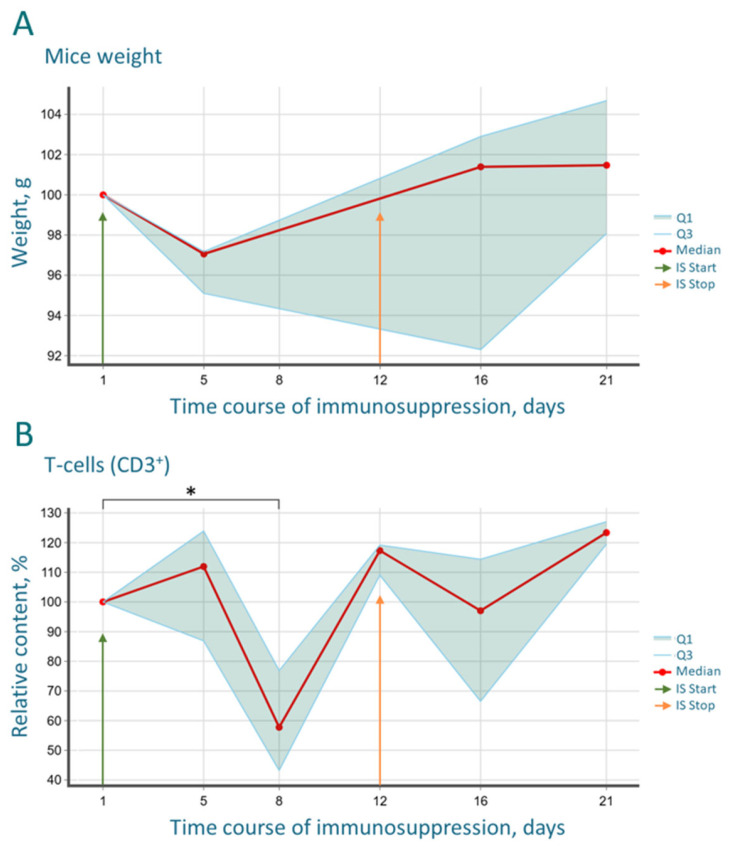
(**A**) Dynamics of mouse body weight during and after immunosuppression. (**B**) Dynamics of the relative T-cell count (CD3^+^), * *p*-value = 0.0277. Arrows indicate the start and end days of the immunosuppressive regimen. The analysis was performed on 6 mice.

**Figure 3 cancers-17-04025-f003:**
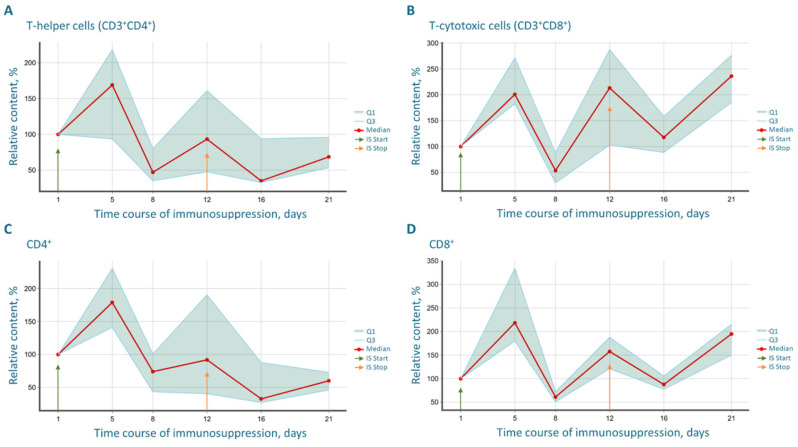
Dynamics of the relative counts of: (**A**)—T-helper cells (CD3^+^CD4^+^); (**B**)—T-cytotoxic cells (CD3^+^CD8^+^); (**C**)—CD4^+^ cells; (**D**)—CD8^+^ cells. Arrows indicate the start and end days of the immunosuppressive regimen. The analysis was performed on 6 mice.

**Figure 4 cancers-17-04025-f004:**
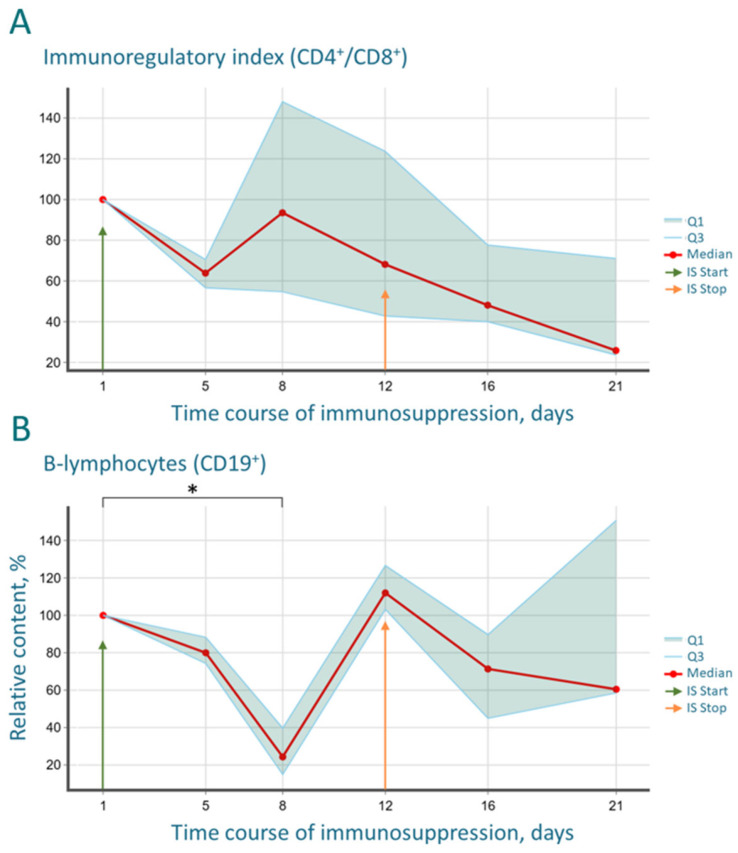
(**A**) Dynamics of the immunoregulatory index (CD4^+^/CD8^+^) during drug-induced immunosuppression. (**B**) Dynamics of the relative B-lymphocyte count. CD19 groups at Day 1 and Day 8 differ statistically significant: adjusted * *p*-value = 0.0277. Arrows indicate the start and end days of the immunosuppressive regimen. The analysis was performed on 6 mice.

**Figure 5 cancers-17-04025-f005:**
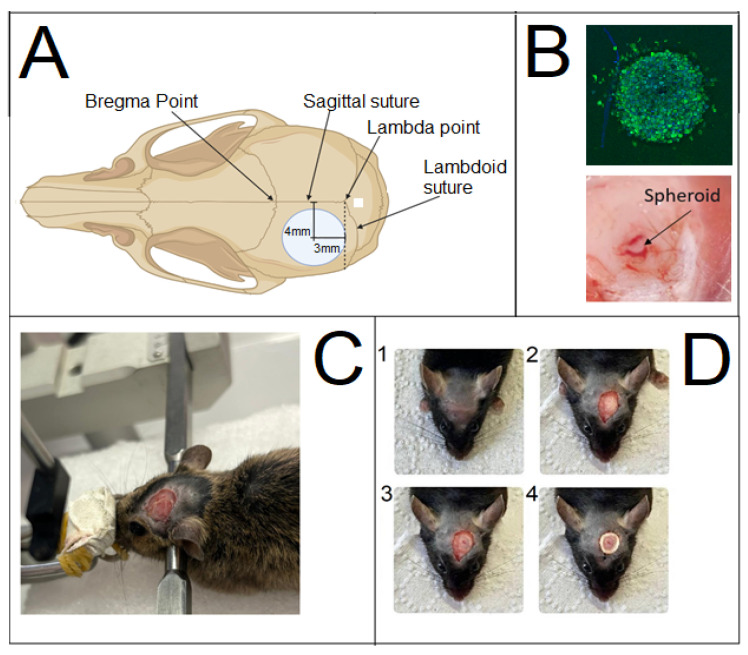
The process of mouse skull trepanation: (**A**)—Schematic representation of the mouse skull with key structures and standard coordinate points (according to the atlas The Mouse Brain in Stereotaxic Coordinates by Paxinos G. and Franklin J. [34]); (**B**)—Spheroid figures prior (confocal fluorescent microscopy imaging: the green fluorescence indicates living cells labeled with Calcein-AM, blue indicates cell nuclei counterstained with DAPI) and right after transplantation (light microscopy); (**C**)—A mouse secured in the stereotaxic apparatus using ear bars and a jaw adapter; (**D**)—Stages of the trepanation procedure: (1)—stage after hair shaving, (2)—stage after skin incision, (3)—stage after creation of the burr hole in the skull, (4)—stage after formation of the cranial window. In all immunosuppressed mice, the xenogeneic cell spheroids successfully engrafted and were visualized for 10 days, until the release of the bioconvertible tracer Calcein from the cells. In the non-immunosuppressed mice, the cell count dropped sharply to 5–7 per field of view by day 3, and the fluorescent signal disappeared completely by day 7. No differences in the tolerability or efficacy of the immunosuppression protocol were observed between the two mouse strains.

**Figure 6 cancers-17-04025-f006:**
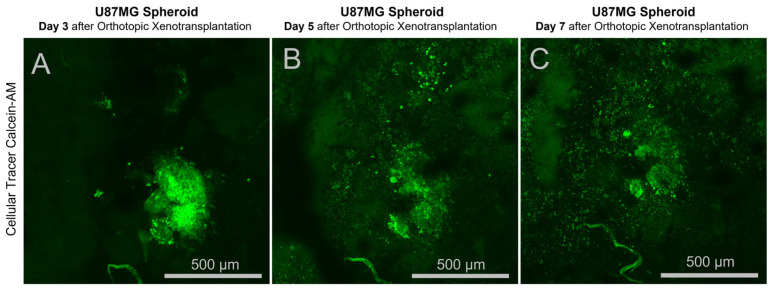
In vivo laser microscopy data from xenograft model animals with induced tumor spheroids of U87MG cells with different genotypes. Intravital fluorescent microscopy through a transcranial optical window.

**Figure 7 cancers-17-04025-f007:**
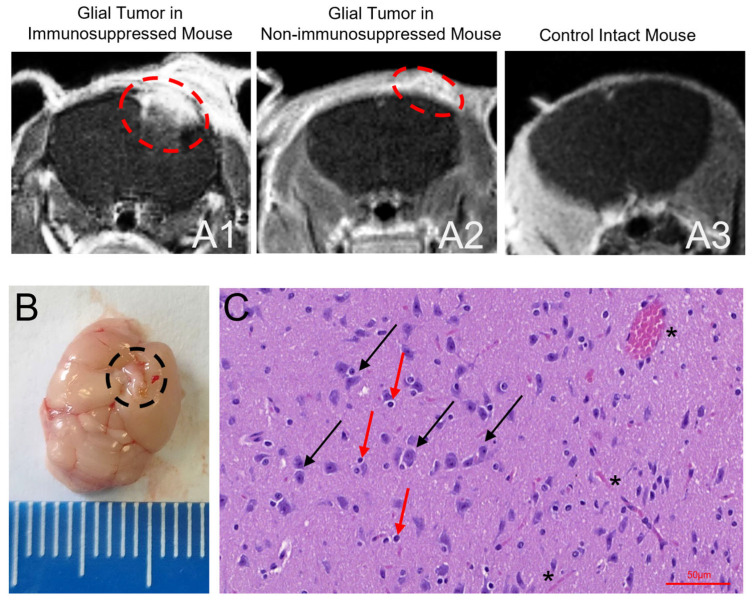
In vivo assessment of glioma growth 20 days after orthotopic transplantation. T1-weighted contrast-enhanced MRI (gadodiamide, Omniscan) is shown. (**A1**) Immunosuppressed mouse with a xenograft demonstrating active tumor growth (red dashed circle indicate tumor). (**A2**) Non-immunosuppressed mouse showing a weak signal at the injection site, indicating a lack of engraftment (red dashed circle indicate tumor). (**A3**) Control mouse (intact brain). (**B**) Autopsy of a mouse brain with a transplanted glial tumor (black dashed circle indicate tumor). (**C**) Histological analysis of the glial tumor. Black arrows show glial tumor cells in brain sections, red arrows indicate lymphocytes and, * indicate blood vessels. Magnification 400×.

**Figure 8 cancers-17-04025-f008:**
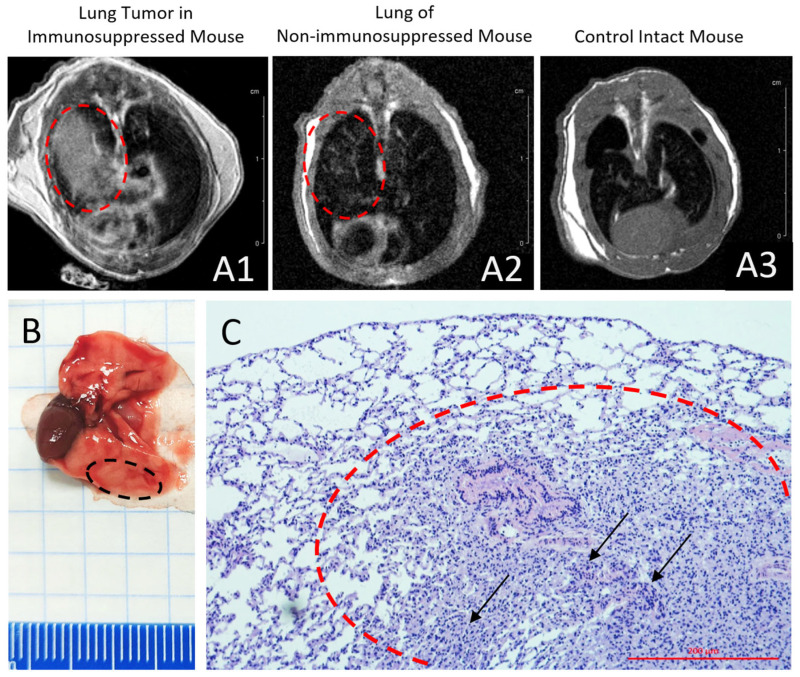
Lung tumor growth 14 days after orthotopic xenotransplantation of primary lung culture. T1-weighted contrast-enhanced MRI (gadodiamide, Omniscan) is shown. (**A1**) Immunosuppressed mouse with a xenograft demonstrating active tumor growth. (**A2**) Non-immunosuppressed mouse showing the injection site, indicating a lack of engraftment. (**A3**) Control mouse (intact lungs). (**B**) Autopsy of mouse lungs with a transplanted glial tumor. (**C**) Histological analysis of the lung tumor. Black arrows show lung cancer cells in lung sections, and dashed circles include tumor areas. Magnification 20×. Scale bar, 200 µm.

**Figure 9 cancers-17-04025-f009:**
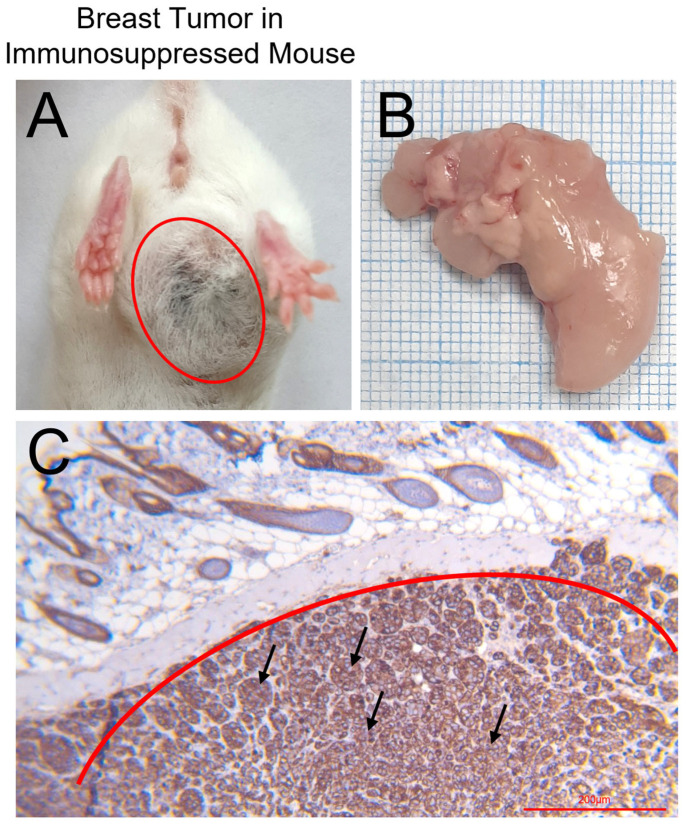
Breast tumor growth three weeks after subcutaneous transplantation of MCF7 cell line. (**A**) A mouse with a transplanted breast tumor. (**B**) Autopsy of a mouse breast tumor. (**C**) Immunohistological analysis of the breast tumor. Black arrows show Her2/neu positive breast tumor cells in tissue sections and solid red circles include tumor areas. Magnification 20×. The successful establishment of these orthotopic and subcutaneous xenograft models in partially immunosuppressed mice provided a valuable source of tumor tissue for subsequent studies. Following humane euthanasia at predefined experimental endpoints, tumor samples were collected and banked for downstream molecular and pharmacological analyses. These biological resources were instrumental in parallel research projects focused on the development and validation of novel tumor-targeting agents, including aptamer-based diagnostic and therapeutic platforms, some results of which have been published [19,20,35,36].

## Data Availability

The original contributions presented in this study are included in the article/Appendix A. Further inquiries can be directed to the corresponding author(s).

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
