# Peer review of "Drug-Induced Partial Immunosuppression for Preclinical Human Tumor Xenograft Models"

_cancers, 2025, doi:10.3390/cancers17244025_

Round 1
Reviewer 1 Report
Comments and Suggestions for Authors
In this work, the authors developed a protocol for pharmacological partial immunosuppression by using a short-term drug regimen to temporarily suppress the immune system for normal mice. The authors validated the models by establishing xenografts of hard-to-treat tumors. My comments related to this work are listed below.
- There is no statistical analysis in this manuscript, they authors should incorporate appropriate statistical methods to support the validity of their findings. Specifically, they should clearly describe the statistical tests used, justify their selection, report sample sizes, and provide measures of variability.
- The representative FCM plots and corresponding gating strategies are not included in the current manuscript. The authors should provide these essential data to ensure transparency and reproducibility of their analyses.
- In Figure 7C, Figure 9C, a scale bar is missing.
- It would be important for the authors to address whether the drug-induced partial immunosuppression has any impact on overall mouse survival or leads to other adverse effects. Specifically, the manuscript should clarify whether the treated mice exhibited increased morbidity, weight loss, altered behavior, susceptibility to infections, or any organ-related toxicities.
Author Response
Dear Reviewers,
Thank you for your careful reading of our manuscript and your valuable comments, which have helped to improve the quality of the work. Our point-by-point responses are provided below. The English language has been edited by the University of Ottawa editing service. All corresponding changes have been highlighted in the revised manuscript.
Reviewer 1
In this work, the authors developed a protocol for pharmacological partial immunosuppression by using a short-term drug regimen to temporarily suppress the immune system for normal mice. The authors validated the models by establishing xenografts of hard-to-treat tumors. My comments related to this work are listed below.
Comment 1.
- There is no statistical analysis in this manuscript, they authors should incorporate appropriate statistical methods to support the validity of their findings. Specifically, they should clearly describe the statistical tests used, justify their selection, report sample sizes, and provide measures of variability.
Response: We fully agree on the importance of rigorous statistical analysis. The following details have been added to the revised manuscript Section 2.4. Statistical Analysis and the corresponding figure legends.
Statistical analysis was performed using the package in the Anaconda Python 3.12.4 environment.
Given the small sample size, the graphs in the immunomonitoring section present medians and
the non-parametric Mann-Whitney U test was used for comparing metrics between two independent time points, justified by the data characteristics and the absence of an assumption of normal distribution.
The Bonferroni correction was applied to account for multiple hypothesis testing in pairwise comparisons.
P-values are indicated in the relevant figure legends.
The cohort size for immunomonitoring (n=6) is clearly stated at the beginning of Section 2.1 and reiterated in the legends for Figures 2-4.
Statistical analysis of transplantation success rates across tumor types revealed the following engraftment efficiencies: 100% for the glial tumor model, 80% for the lung cancer model, and 100% for the MCF7 breast cancer model, highlighting the protocol's efficacy and the inherent variability between tumor types and patient-derived samples. Mice with the tumors were utilized in parallel research projects focused on the development and validation of novel tumor-targeting agents, including aptamer-based diagnostic and therapeutic platforms, some results of which have been published. The references have been added.
Comment 2.
- The representative FCM plots and corresponding gating strategies are not included in the current manuscript. The authors should provide these essential data to ensure transparency and reproducibility of their analyses.
Response: We thank the reviewer for this pertinent comment. A detailed plot illustrating the representative gating strategy for lymphocyte subpopulation analysis has been added to the manuscript as Supplementary Figure S1. This ensures full transparency and reproducibility of our analysis.
Comment 3.
- In Figure 7C, Figure 9C, a scale bar is missing.
Response: Scale bars have been added to the specified figures. We apologize for this oversight.
Comment 4.
- It would be important for the authors to address whether the drug-induced partial immunosuppression has any impact on overall mouse survival or leads to other adverse effects. Specifically, the manuscript should clarify whether the treated mice exhibited increased morbidity, weight loss, altered behavior, susceptibility to infections, or any organ-related toxicities.
Response: This is a crucial aspect of the protocol safety profile, because the dugs are toxic and influence directly on the immune system. A detailed analysis of side effects has been added to Section 3.1.
Survival: No animal mortality was recorded throughout the entire experiment (lethality = 0%).
Clinical Signs: Characteristic symptoms (lethargy, poor coat condition, reduced food and water intake) were described, peaking around day 5.
Body Weight: The dynamics of body weight, a key indicator of overall condition, are detailed and illustrated in Figure 2A. Weight loss was recorded by day 5, followed by complete recovery, indicating the transient nature of the toxicity.
Body Temperature: Following the reviewer's comment, we have added the core body temperature data to the manuscript (Supplementary Figure S2). While initially regarded as secondary to the primary immunological analysis, this metric offers important systemic physiological context. The recorded decrease (to 37.2°C by day 5) and subsequent rebound hyperthermia (to 38.6°C by day 21) align precisely with the phases of immunosuppression and recovery, strengthening the overall assessment of the protocol's effects.
Infections: Due to the prophylactic use of ketoconazole, and individually ventilated cages no cases of opportunistic mycoses or other overt infections were recorded.
Reviewer 2 Report
Comments and Suggestions for Authors
In this manuscript Gorbushin et al describe the use of an immunosuppressive drug regime to allow Xenograft implantation in an immunocompetent mouse model.
The background is well structured and informative and provides relevant citations.
The novelty of the study is average as the regime has been previously described by Jivrajani et al, however the testing of multiple tumor types and use of orthotopic models adds additional value.
Figures are clear and diagrams explain concepts well. Please add number of animals used into each figure legend, especially 2-4. It is not clear what the value of 3C and 3D are. In fig 4A is there any significant difference. Please add statistical tests used in fig legends where relevant.
What controls are used in fig 7A2, 8A2? Please specify. It would be useful to have data on tumor growth in mice that did not receive the immunosuppressive conditioning to illustrate that the tumors don’t grow without the immunosuppression.
The number of mice used in the study is low, from what I understand only 1 mouse was used for each tumor studied. I understand that using more mice would be costly and potentially wasteful however I wanted to point out this limitation.
Overall good manuscript that provides useful insights into mouse models to be used in future studies.
Author Response
Dear Reviewers,
Thank you for your careful reading of our manuscript and your valuable comments, which have helped to improve the quality of the work. Our point-by-point responses are provided below. The English language has been edited by the University of Ottawa editing service. All corresponding changes have been highlighted in the revised manuscript.
Reviewer 2.
In this manuscript Gorbushin et al describe the use of an immunosuppressive drug regime to allow Xenograft implantation in an immunocompetent mouse model.
The background is well structured and informative and provides relevant citations.
The novelty of the study is average as the regime has been previously described by Jivrajani et al, however the testing of multiple tumor types and use of orthotopic models adds additional value.
Comment 1.
Figures are clear and diagrams explain concepts well. Please add number of animals used into each figure legend, especially 2-4. It is not clear what the value of 3C and 3D are. In fig 4A is there any significant difference. Please add statistical tests used in fig legends where relevant.
Response: The information (n=6) has been added to the legends for all Figures 2, 3, and 4.
Figures 3C and 3D: These graphs show the dynamics of the relative counts of all CD4⁺ and CD8⁺ cells, respectively (including both CD3⁺ and potentially CD3⁻ populations). Their inclusion allows for a more comprehensive view of changes in these key lymphocyte lineages under therapy and confirms that the patterns we observed for the CD3⁺CD4⁺ and CD3⁺CD8⁺ subpopulations (Figs. 3A, 3B) also hold true for the broader pool of cells bearing these markers.
Figure 4A (CD4/CD8 Index): You are correct; despite a clear downward trend, the differences between individual time points did not reach statistical significance after Bonferroni correction. This is reflected in the updated figure legend.
The exact p-values for significant comparisons are indicated on the 2-4.
Comment 2.
What controls are used in fig 7A2, 8A2? Please specify. It would be useful to have data on tumor growth in mice that did not receive the immunosuppressive conditioning to illustrate that the tumors don’t grow without the immunosuppression.
Response: We thank the reviewer for pointing this out. Corresponding information has been added to the revised text.
We fully agree on the importance of this control. While we initially did not include these data in the first submission, considering them less informative for the primary aim of characterizing the new protocol, we now understand and acknowledge their importance. Consistent with our prior experience orthotopic xenografts from human primary cultures or cell lines do not engraft or form tumors in immunocompetent mice without any immunosuppression. In this study, we focused on characterizing the new protocol and demonstrating its efficacy for tumor engraftment. Representative images of control mice are now provided.
Comment 3.
The number of mice used in the study is low, from what I understand only 1 mouse was used for each tumor studied. I understand that using more mice would be costly and potentially wasteful however I wanted to point out this limitation.
Response: We thank the reviewer for this fair comment. Our initial scope was to demonstrate the immunosuppression and transplantation methodologies themselves. All mice with tumors from this foundational study were utilized in subsequent, separate experimental projects. In total, the methodology described here has been successfully used to generate cohorts for follow-up studies: over 200 mice with orthotopic glioma xenografts, approximately 50 mice with patient-derived lung cancer xenografts, and about 40 mice bearing subcutaneous MCF7 tumors. Data from these subsequent investigations are being prepared for separate publications, which will focus on therapeutic or mechanistic findings and thus will not include a detailed redescription of the foundational model generation protocols presented here. Where results have already been published, the relevant citations have been added to this manuscript. This demonstrates the robustness and reproducibility of the model across different experiments and research groups.
Comment 4.
Overall good manuscript that provides useful insights into mouse models to be used in future studies.
Once again, we thank you for your constructive criticism, which has significantly strengthened the rigor and clarity of our manuscript. We hope that the revisions and additions satisfactorily address all the points raised.
Round 2
Reviewer 1 Report
Comments and Suggestions for Authors
The authors have addressed most of the reviewers' concerns.